# Lessons learnt for digital inclusion in underserved communities from implementing a covid virtual ward

Rosanna Fox[1], Zeshan Saeed[1], Sadia Khan[2]*, Harry Robertson[1], Sophie Crisford[1], Andrew Wiggam[1], Abby Foley[1], Farhana Raza[1], Michael Wright[1]

**1** West Middlesex University Hospital, Twickenham Road, Isleworth, United Kingdom, **2** Imperial College London, Exhibition Road, London, United Kingdom

* snkhan@imperial.ac.uk

## Abstract

The factors associated with digital exclusion in the covid virtual ward population at a North West London teaching hospital were assessed in this study. Patients discharged from the covid virtual ward were contacted to give their feedback on their experience. Questions were tailored to whether or not the patient used the Huma app during their time on the virtual ward and were subsequently divided into 'app user' and 'non-app user' cohorts. The non-app user population accounted for 31.5% of the total patients referred to the virtual ward. Four major themes drove digital exclusion in this group: language barriers, access, inadequate information/training, and poor IT skills. In conclusion, incorporating additional languages and improving hospital-setting demonstration and information provision to patients prior to discharge were highlighted as key factors for reducing digital exclusion in the covid virtual ward patients.

## Introduction

The Covid-19 pandemic led to a rapid deployment of digital technologies into healthcare pathways, often at speeds that would have been unthinkable in a pre-pandemic era as healthcare systems tried to provide as much care as possible whilst minimising risks of nosocomial coronavirus infection. One such digital innovation was the Covid-19 virtual ward (CVW) which, after being shown to be successful in the first wave of the pandemic, was supported centrally by the National Health Service to be rolled out more widely. Our institution provides hospital based healthcare services for an economically deprived and ethnically diverse area of North West London–a region that saw high case numbers of patients hospitalised due to coronavirus in the timeframe of the study. We undertook rapid deployment of a CVW to care for patients at the start of the second wave of the pandemic (December 2020 until April 2021). The deployment of the CVW used a commercially available digitally platform (Huma remote patient monitoring solutions—comprising a smartphone app with a clinician dashboard) combined with a small pulse oximeter and a written information sheet giving a telephone number for the service.

**Funding:** The authors received no specific funding for this work.

**Competing interests:** The authors have declared that no competing interests exist.

## What is the Covid-19 virtual ward?

The 'Covid-19 virtual ward' (CVW) was set up at West Middlesex University Hospital (WMUH) in January 2021 in response to the UK's winter peak of the Covid-19 pandemic. This was designed to limit in-hospital spread, free up hospital beds and reduce strain on the NHS and has since been widely successful across the UK [1]. Due to the unprecedented pressures of bed capacity and rapidly increasing admissions of critically unwell patients requiring supplementary oxygen or ventilatory support the trust was required to manage certain patients externally. Patients requiring ongoing monitoring, but with no immediate need for supplementary oxygen support were therefore monitored remotely via a smart phone app and telephone consultation. Patients deemed unsafe at home, or in need of clinical review and/or possible oxygen therapy were asked to urgently return to hospital. Some required ambulance transfer. Between December 2020 and February 2021, the CVW managed a total of 142 patients. Readmission rates from the CVW back to hospital were 18.3% for this period. Referrals were made by both ward and emergency department (ED) teams.

## What is digital exclusion?

Digital exclusion refers to the barriers preventing effective use of digital technology. It is a term typically used in reference to social, economic and educational development. Many factors have been suggested to affect the access to and uptake of effective digital technology use across populations including: age, sex, ethnicity, education, household income, physical and mental health, information technology (IT) literacy, and rural living [2–7]. However, the literature is not extensive and in particular there remains very little research into digital exclusion when IT is used in health management and development.

## Why is digital exclusion of interest to healthcare professionals?

Widespread ability to access and effectively use digital technology has grown exponentially over the last decades. This has offered new and exciting opportunities to the medical profession in terms of remote/digital monitoring of and access to patients. Nationwide, we have been seeing a move towards increased use digital technology monitoring in the outpatient setting, for instance FreeStyle Libre devices in diabetes management and AliveCor Kardia in atrial fibrillation [8,9]. However interest in outpatient monitoring accelerated dramatically in response to the pandemic [10].

However, the 'digital exclusion' of certain patient groups (e.g. the elderly, lower socioeconomic, ethnic minorities or those with chronic physical and mental health conditions) may inhibit or even prevent their access to such healthcare provision and also lead to further health inequalities that the covid pandemic has brought into sharp focus. Prior to the pandemic, there had been a move towards virtual consultations in NHS general practice, which has been taken up in earnest with new covid restrictions in place to protect both staff and patients and triaging taking place via telephone to limit contact as appropriate. With this practice becoming more widespread, and the possibility of digital healthcare expanding into other areas of outpatient medicine, we feel strongly any barriers to uptake should be addressed now and avoid future preventable healthcare inequality especially in underserved communities.

This study aims to review the factors associated with digital exclusion in patients under the care of the CVW at WMUH in early 2021. If we can elucidate these factors we may be able to mitigate and/or enable underserved groups of patients who may otherwise be unable or unqualified to access such systems of healthcare in the future. Given the possible scope of digital patient care, it is vital that we are able to recognise patients at risk of digital exclusion and support them appropriately.

## Methodology

Patients aged 18 years or more discharged by WMUH that had been diagnosed with or received treatment for confirmed or suspected Covid-19 infection were potential candidates for the CVW.

Further criteria for virtual ward admission are listed below:

- Maintained SpO2 > = 92% on room air

- <3% drop in SpO2 on ambulation (ED-discharges)

- Afebrile for 48 hours (ward-discharges)

- Down-trending CRP for ward-discharges; CRP <60 mg/L for ED-discharges

- Platelet count > 100 x $10^9$/L

- English speaking (or patients with an English speaker at home with them)

- Access to and ability to use a smart phone and the Huma app

Patients fulfilling the above criteria were referred via email to the CVW by a doctor from their discharging team (emergency department or ward). They were each given a pulse oximeter and written instructions to download the Huma app. The CVW team (consisting of two doctors and two nurses) would contact each referred patient within 24 hours of discharge from the hospital. Further telephone assessments were then carried out on days 3, 5, 7 and 10 post-discharge and patient observations (heart rate, oxygen saturations, core temperature and self-reported symptoms) were remotely monitored via the app. Any patients with oxygen saturations recorded at 92% or below, or with other 'red flag' observations e.g. persistent high core temperature (>38 degrees Celsius), significant tachycardia (>120 beats per minute), or self-reported significant worsening of symptoms were contacted urgently. In general, those patients with worsening symptoms and oxygen saturations persistently at 92%, or any patient with recorded oxygen saturations below 92% were advised to urgently re-present to WMUH emergency department. The CVW team also offered a Covid-19 hotline service, operating 8am-8pm daily. Patients on the CVW were encouraged to contact this number if they had any concerns about their chest symptoms or noticed a new drop in their pulse oximeter saturations.

The majority of patients were discharged from the CVW at day 10 (excluding the cohort that required readmission). Those patients successfully discharged were subsequently contacted via phone to give their feedback on their CVW experience via a structured interview. Questions were grouped into themes around 1. Patient satisfaction, 2. Digital exclusion and 3. Recovery pattern. In terms of theme 2, patients were asked to clarify their age and first language and to answer questions surrounding the information given to them prior to discharge regarding pulse oximeter and app use, and any issues they encountered. Patients were divided into 'app users'—those who had successfully downloaded the Huma app and were regularly inputting data (at least twice daily), and 'non-app users'—who had either failed to download the app, or had not input any data. Questions were tailored as shown in the index; 'app users' were asked the questions detailed in S1 Index and 'non-app users' were asked the questions detailed in S2 Index. Adverse outcomes were listed for any patients requiring readmission or who died. Non-app users were asked to give their ethnicity. Age for all patients was collected from the trust's digital health records. Questions were asked in a close-ended format, except questions 2–4 for non-app users which were open-ended. App users were encouraged to give additional comments in the last question (14) of their survey, otherwise questions were strictly yes/no or scaled strongly agree to strongly disagree.

## Results

Overall the ward managed 142 patients during the December 2020 –February 2021 period. 97 (67.8%) patients input data to the app and were considered 'app users'. 45 (31.5) did not input data to the app or did not download it and were considered 'non-app users'. 4 (2.8%) patients passed away. Of the 97 app user patients, 18 (18.5%) had adverse outcomes, with 1 death. Of the 45 non-app users, 12 (26.7%) had adverse outcomes, with 3 deaths. Adverse outcomes in the non-app user groups were not significantly higher (p value 0.27, Chi-square statistic 1.21).

Data was collected for 65 patients that used the app. 2 patients did not wish to take part, with 29 patients not answering multiple calls or being unreachable. The mean age of the app user cohort was 50.1. There were 45 patients referred to the CVW that did not use the app. Data was collected for 23 patients. 2 patients declined to take part in the survey, 1 patient never received a pulse oximeter, 14 were unreachable and 1 had no number recorded. The mean age of the non-app user cohort was 55.8 (age range 26–87). We had anticipated advanced age to be a significant factor and indeed, the mean age of the non-app user population was significantly higher than that of the app user population (p value = 0.03).

Overall comments were extremely positive form the app using group. Amongst patients that suffered from an adverse outcome a common theme in additional comments was that the app saved their life and they were extremely appreciative of the team keeping in constant communication with them. However, it was highlighted that 11 patients did not use the app themselves but required a family member to input and record the data on their behalf. Of app users, 71% reported this directly affected their decision to re-attend hospital or not. 89% of app users found the app easy to use with 92% reporting the CVW set-up made them or their family feel reassured.

Non-app users mostly commented on language barriers or inadequate demonstration or explanation of the pulse oximeter and/or the app in the hospital setting prior to discharge. One patient was illiterate and another reported never receiving a pulse oximeter. Fig 1 is used to demonstrate this graphically, with phone and internet access being shown as clear barriers to app use. Four major themes were highlighted by the non-app using cohort.

### 1. Language barriers

The non-app user cohort was ethnically diverse and some of these patients did not speak or read English. 48% of the cohort reported an app in another language would have helped.

### 2. IT Skills

26% of non-app users reported difficulties with the app use. None reported difficulties using the pulse oximeter.

### 3. Information and training

48% of non-app users reported training would have helped them use the app appropriately.

### 4. Access

4% of non-app user patients did not have a phone, 9% did not have internet service, and 35% did not have smartphone technology and therefore no ability to download or use apps. 4% of the non-app user cohort reported disability affected their ability to use the app.

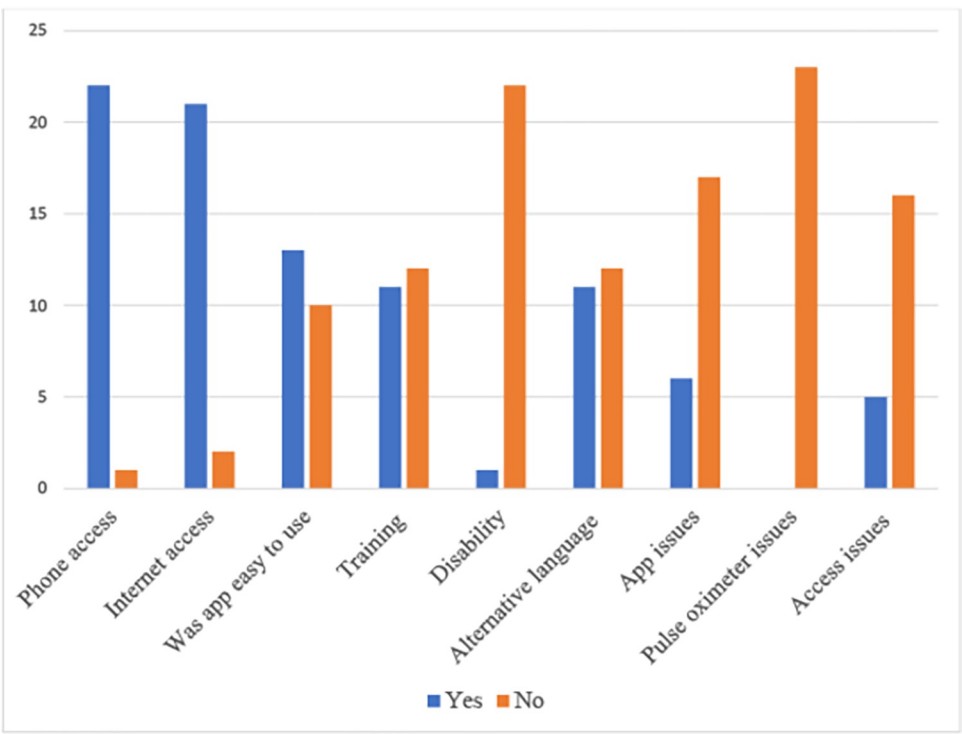

**Fig 1. Graphical representation of the non-app users answers to questions 5–9 (detailed in S2 Index).**

## Discussion

The CVW aimed to reduce both mortality and hospital stay. Ultimately, 142 patients that would have otherwise been admitted/had a longer stay in hospital were discharged home; reducing hospital workload. Interestingly, the adverse outcomes were not significantly different between cohorts, however the overall patient numbers in this study are small and this may in part reflect that patients selected for a virtual ward are not as unwell.

The CVW was largely successful with patients and their families overall giving very positive feedback and overwhelmingly feeling reassured going home with the pulse oximeter. Indeed, over 90% of the app user cohort reported the CVW was reassuring for either themselves of their family. The additional comments given by patients in this study highlighted four key issues in the non-app user cohort; language barriers, IT skills, inadequate information/training, and access to digital technology. Other socioecomonic factors e.g. literacy should be taken into account but the effect of these factors was less than expected on the non-app using cohort with only one patient reporting illiteracy. Nevertheless, these patients are at high risk of digital exclusion and should be identified early to prevent adverse outcomes.

In light of these factors, we would focus in the future on providing additional support for those patients with limited IT skills or experience to ensure this does not prevent their ability to access digital medical services in the community. App registration requiring codes and additional logins should be undertaken in hospital to ensure patients are discharged safely. For those patients without phone/internet/app access and therefore unable to partake in digital monitoring, provisions were made by the CVW team during the pandemic e.g. recording oxygen saturations manually and texting the results to the CVW hotline phone. However, this bought up several safety issues and relied heavily on patient cooperation, and on a larger scale it would be impossible to manually monitor so many patients.

Ultimately, many patients were inappropriately referred to the CVW, given the criteria for admission required internet and smart phone access and English spoken by at least one member of the household. Those patients not meeting referral criteria were all in the non-app user group which serves to highlight the importance of discharge planning where remote monitoring is required.

Although our study demonstrated clear factors associated with digital exclusion in the non-app using cohort, it presents inherent limitations. Firstly, the questionnaires used for patient feedback are not validated instruments to assess health related digital technologies. Additionally, the questions used mostly capture subjective patient impressions rather than offering a reliable measure of e.g whether pre-discharge hospital-based app training was sufficient or necessary. Secondly, as suggested above, many of the non-app users were inappropriately referred to the CVW and did not meet the initial referral criteria. The criteria for referral sought to minimise the risk of digital exclusion e.g. "must be English speaking (or patients with an English speaker at home with them)" and "must have access to and ability to use a smart phone and the Huma app". In cases where these initial criteria were not met, the resulting feedback from the patient is likely biased towards digital exclusion, as these patients are unlikely to be successful app users. Thirdly, telephone interviews were conducted with a different set of questions for app users and non-app users which allowed for useful descriptive feedback through open ended questions but unfortunately limited direct comparison between groups. App users were not asked to clarify their ethnicity which prevents direct comparison of this variable between the two cohorts. Additionally, patients in both cohorts were not asked their socio-economic status which would have been a useful comparator.

There is currently no other study in the wider literature looking specifically at digital exclusion in remotely monitored patient populations. In fact, there is a relative paucity of literature investigating factors driving digital exclusion or its relevance in the healthcare setting. Given this clear gap in the current research we feel our study provides an excellent foundation for further research into this challenging and highly relevant area as we move towards and increasingly digital space in healthcare.

Future research should seek to use a validated objective tool to assess the possible causes of digital exclusion in order to direct resources appropriately to support 'at risk' populations and target patients at risk of digital exclusion for factors such as age, IT skills, language barriers, health issues or deprivation. Our work has highlighted the need for additional languages and IT training in those patients considered higher risk of digital exclusion. Staff training and involvement in pathway design would also help to limit the numbers of non-app users in the future and flag those that may struggle in an outpatient setting to manage digital technology.

## Conclusion

This study highlights four major themes driving digital exclusion: language barriers, access, inadequate information/training, and poor IT skills. Almost half of the non-app user cohort felt additional languages and training would have enabled app use. Advanced age was predictive of digital exclusion in our CVW population. As a result, we will seek to incorporate additional languages and improve hospital-setting IT training for those patients being remotely monitored using digital platforms in the future. Clearly, further study in this area would be useful to direct resources and support patients appropriately.

As healthcare moves towards a more digital space, it is crucial we use the lessons of the pandemic to safeguard patients at risk of digital exclusion. Inclusion of other languages and cultures in both hospital-based patient training and home-based app use could protect a significant proportion of these high risk individuals and enable them to access and benefit from digital healthcare.

## Supporting information

**S1 Index. Formal interview questions asked over phone to app-users (defined as those who has downloaded the Huma app and were regularly inputting data at least twice daily).**
(DOCX)

**S2 Index. Formal interview questions asked over phone to non-app used (defined as those who either did not download the Huma app, or those who did not input data).**
(DOCX)

**S1 Data. Anonymised Data.**
(XLSX)

## Author Contributions

**Conceptualization:** Sadia Khan.

**Data curation:** Zeshan Saeed, Harry Robertson, Sophie Crisford, Andrew Wiggam.

**Formal analysis:** Rosanna Fox.

**Investigation:** Zeshan Saeed, Harry Robertson, Sophie Crisford, Andrew Wiggam, Abby Foley, Farhana Raza.

**Methodology:** Rosanna Fox, Abby Foley, Farhana Raza, Michael Wright.

**Project administration:** Michael Wright.

**Supervision:** Zeshan Saeed, Sadia Khan.

**Writing – original draft:** Rosanna Fox.

**Writing – review & editing:** Rosanna Fox, Sadia Khan.

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
