## [Decision Letter · Decision Letter 0]

1 Sep 2022

PDIG-D-22-00165

Title - Digital Exclusion in the covid virtual ward at a North West London Hospital: lessons learned.

PLOS Digital Health

Dear Dr. Khan,

Thank you for submitting your manuscript to PLOS Digital Health. After careful consideration and based on reviewer comments, we feel that it has merit but does not fully meet PLOS Digital Health's publication threshold as it currently stands. Therefore, we invite you to submit a revised version of the manuscript that addresses the points raised during the review process.

Please submit your revised manuscript within 60 days (due Oct 31st 2022). If you will need more time than this to complete your revisions, please reply to this message or contact the journal office at digitalhealth@plos.org. Please include the following items when submitting your revised manuscript:

We look forward to receiving your revised manuscript.

Kind regards,

Rutwik Shah, MD

Guest Editor

PLOS Digital Health

Journal Requirements:

1. Please amend your online Financial Disclosure statement. If you did not receive any funding for this study, please simply state: “The authors received no specific funding for this work.”

2. Please update your online Competing Interests statement. If you have no competing interests to declare, please state: “The authors have declared that no competing interests exist.”

3. Please note that your Data Availability Statement is currently missing the DOI/accession number of each dataset. If your manuscript is accepted for publication, you will be asked to provide these details on a very short timeline. We therefore suggest that you provide this information now, though we will not hold up the peer review process if you are unable.

4. Please provide separate figure files in .tif or .eps format only and remove any figures embedded in your manuscript file. Please also ensure that all files are under our size limit of 10MB.

For more information about how to convert your figure files please see our guidelines: https://journals.plos.org/digitalhealth/s/figures

5. Please ensure that you refer to Figures 1, 3, and 4 in your text as, if accepted, production will need these references to link the reader to the figures.

6. We notice that your supplementary materials (Indexes 1 and 2) are included in the manuscript file. Please remove them and upload them with the file type 'Supporting Information'. Please ensure that each Supporting Information file has a legend listed in the manuscript after the references list.

7. We have noticed that you have uploaded Supporting Information files, but you have not included a list of legends. Please add a full list of legends for your Supporting Information files after the references list.

Additional Editor Comments (if provided):

Reviewers' comments:

Reviewer's Responses to Questions

**Comments to the Author**

1. Does this manuscript meet PLOS Digital Health’s publication criteria? Is the manuscript technically sound, and do the data support the conclusions? The manuscript must describe methodologically and ethically rigorous research with conclusions that are appropriately drawn based on the data presented.

Reviewer #1: Partly

Reviewer #2: Yes

2. Has the statistical analysis been performed appropriately and rigorously?

Reviewer #1: N/A

Reviewer #2: No

3. Have the authors made all data underlying the findings in their manuscript fully available (please refer to the Data Availability Statement at the start of the manuscript PDF file)?

Reviewer #1: No

Reviewer #2: No

4. Is the manuscript presented in an intelligible fashion and written in standard English?

Reviewer #1: Yes

Reviewer #2: Yes

5. Review Comments to the Author

Reviewer #1: The paper addresses a very important instance of digital transformation in healthcare. It is timely and potentially very useful for researchers and developers in this domain. 

Nevertheless, I'd like to point out some methodological issues that the authors should address as they revise the manuscript: 

- Important methodological details are missing: What instrument did you use (a survey, a survey + interviews, a survey + room for open comments, survey + data)? 

- Provide details on ethics approval and informed consent for data collection, data use and analysis. How did you get age-related info from the app users if you did not ask that in Index1? Did you have authorisation to use such information, say, from health records for research purposes? 

- Since you did not ask app users about their ethnicity, results in figure 2 are not really informative. - How did you analyse data form open comments in Index1? (see table 2) 

- Is question 9 in Index2 an open comment question or a y/n question? 

- How comes you did not ask about socio-educational status, given you rightly state this is a major determinant of digital exclusion? In the absence of this more granular information it is hard to make sense of, for instance, answers regarding ease-of-use. 

- How comes proficiency in English, access to and ability to use apps are inclusion criteria, and some respondents ended up not being able/willing to use the app and having problems with English? This is hard to make sense of. 

- Question 5.c in Index2 seems a y/n question, while question 13 in Index 1 (on the same topic) uses a scale. It is thus hard to interpret the results in this case. 

- Important limitations to specify in the paper: the authors need to specify that the questionnaires they have used are not validated instruments to assess users acceptance of health-related digital technologies. 

- Furthermore, the questions are framed in such a way that they capture patients subjective impressions, rather than being a reliable measure of, e.g., whether or not enough information was provided or training was needed. 

- Discussion: this section is not actually a discussion, but rather a presentation of other results (that should go in the results section instead) followed by commentary. I'd expect this section to discuss the results in light of similar studies and relevant literature in the the field of digital health and digital health ethics. 

Other minor points: 

- I wouldn't say "in both surges" in the abstract as there have been more than two waves of COVID19 in the UK. Add "in the timeframe of the study" or something of the like. 

- Did CVW admit 142 or 143 patients? See lines 53 and 120. 

- I would not use the expressions "digitally included" and "digitally excluded" as they carry an evaluative connotation. Use "app users" and "app non-user" instead.

Reviewer #2: Review for Digital Exclusion in the covid virtual ward at a North West London Hospital: lessons learned.

Overall, this is an important paper in a topical area. The paper has the potential to make huge contributions to the area of health inequities and digital inclusion. However, more needs to be done to clarify aims, objectives and results and discussion sections need considerable work. 

I wonder if the title could be changed to highlight the novel aspects of the study. For example: “Lessons learnt for digital inclusion in marginalised/underserved communities (use most appropriate term here and keep consistent throughout the write up) from implementing a covid virtual ward” 

Abstract:

• Objective: I would recommend changing “causing” to “that lead to” or “factors associated with” or “drivers of” causality not determined via your study, rather associations and differences noted. Another way to re-work the objectives would be to replace “factors causing digital exclusion in patients” could be changed to “facilitators and barriers associated with uptake of remove digital monitoring in the outpatient setting (or at home) during covid-19 pandemic in economically deprived and/or ethnically diverse groups”

Intro:

Overall, this section is well written and introduces key issues in a concise manner. I would recommend the following minor edits

• Line 35: economically “deprived”

• Line 48 replace “forced” with “required to”

• Line 70, specify with groups

• Line 68-69, give examples of remote/digital monitoring in the outpatient setting

• Line 80 – what do you mean by vulnerable groups? Are you referring to underserved communities or economically deprived and/or ethnically diverse groups? Might be worth ensuring that terminology is kept consistent throughout the paper

• Line 77-78 – revise objective of the study as suggested above.

Method:

A few further details required for clarity. 

• Line 85-86 – how many patients were assessed for eligibility and how many were finally enrolled in the study.

• By design the eligibility criteria excludes some groups – such as those unable to speak English or having a carer/family member who speaks English or have access to a smart phone or be willing to download the Huma app. Was data collected on how many were excluded based on these criteria? It might also be worth reflecting on this in the discussion.

• Line 95: were other methods considered/used to refer patents into CVW– for example telephone rather than the exclusive use of emails to refer patients to the CVW? It might be worth also reflecting on whether the method of enrolling into the study itself might have been digitally biased and excluded those who might experience barriers to accessing digital technologies. 

• Line 111: was feedback collected as a survey? Who conducted this? How was feedback collected? Online, via the app, on the phone? Were questions open ended, closed ended, some example questions – more details required here.

• Lines 114-117: how were classifications made on whether users of CVW were “app users” or “non app users”. Who made this classification, how and when was it done? I thought having the HUMA app was an inclusion criterion (see Line 94). My understanding then is that “non app users” are patents who had access/had downloaded app but did not use the app.

Results:

This section needs work. I would suggest having clear research questions and then reorganising your result to include means, and appropriate statistical tests to show significant differences between the groups and highlight any interesting results. 

• Table 1 and Figure 1 do not contribute to the understanding of the topic or add value to the result section. 

• I would recommend starting with a section describing the participants characteristics of both groups– including ethnic origin of participants (Figure 2).

• I would then present the results as means (SDs) and then move on to do appropriate test (t-tests) to show difference between groups. 

• Figure 4 results interesting and most relevant. However no real reporting of the results in the text. Can this be elaborated and some more appropriate tests conducted. 

Discussion:

• This section reads more as a summary of the results. I would have liked to see some further engagement with the wider literature in the field. There is potential here to highlight the implications this work has on the lessons for digital inclusion in underserved/marginalised communities. 

• Could this be added to this section. Have other studies evaluated CVW and digital exclusion? Or can the current results be compared with wider studies on digital exclusion?

• What contributions does the current research make to the field?

• What were some limitations of the current study? 

• Are there any further lessons that can be learnt for implementation in practice?

• Do the authors have any suggestions for future studies in the area?

6. PLOS authors have the option to publish the peer review history of their article (what does this mean?). If published, this will include your full peer review and any attached files.

**Do you want your identity to be public for this peer review?** For information about this choice, including consent withdrawal, please see our Privacy Policy.

Reviewer #1: No

Reviewer #2: No

---

## [Editor Report · Decision Letter 1]

18 Oct 2022

Lessons learnt for digital inclusion in underserved communities from implementing a covid virtual ward.

PDIG-D-22-00165R1

Dear Dr. Khan,

We are pleased to inform you that your manuscript 'Lessons learnt for digital inclusion in underserved communities from implementing a covid virtual ward.' has been provisionally accepted for publication in PLOS Digital Health.

Best regards,

Rutwik Shah, MD

Guest Editor

PLOS Digital Health
